# Role of Epiregulin in Lung Tumorigenesis and Therapeutic Resistance

**DOI:** 10.3390/cancers16040710

**Published:** 2024-02-07

**Authors:** Noriaki Sunaga, Yosuke Miura, Tomomi Masuda, Reiko Sakurai

**Affiliations:** 1Department of Respiratory Medicine, Gunma University Graduate School of Medicine, 3-39-15 Showa-Machi, Maebashi 371-8511, Gunma, Japan; miura04@gunma-u.ac.jp (Y.M.); masudat@gunma-u.ac.jp (T.M.); 2Oncology Center, Gunma University Hospital, 3-39-15 Showa-Machi, Maebashi 371-8511, Gunma, Japan; rsakurai@gunma-u.ac.jp

**Keywords:** epiregulin, non-small cell lung cancer, oncogene

## Abstract

**Simple Summary:**

Epiregulin (EREG) is a member of the ErbB family of ligands that plays multiple roles in cellular processes, including cell proliferation, invasion, and angiogenesis. Accumulating evidence has indicated that EREG is involved in lung tumorigenesis and therapeutic resistance. It is becoming evident that EREG contributes to the epithelial–mesenchymal transition, cancer stemness, immune evasion, and resistance to anticancer drugs in several human cancers, including non-small cell lung cancer. In this review, we summarized the current understanding of EREG as an oncogene and discussed its oncogenic role in lung tumorigenesis and therapeutic resistance.

**Abstract:**

Epidermal growth factor (EGF) signaling regulates multiple cellular processes and plays an essential role in tumorigenesis. Epiregulin (EREG), a member of the EGF family, binds to the epidermal growth factor receptor (EGFR) and ErbB4, and it stimulates EGFR-related downstream pathways. Increasing evidence indicates that both the aberrant expression and oncogenic function of EREG play pivotal roles in tumor development in many human cancers, including non-small cell lung cancer (NSCLC). EREG overexpression is induced by activating mutations in the *EGFR*, *KRAS*, and *BRAF* and contributes to the aggressive phenotypes of NSCLC with oncogenic drivers. Recent studies have elucidated the roles of EREG in a tumor microenvironment, including the epithelial–mesenchymal transition, angiogenesis, immune evasion, and resistance to anticancer therapy. In this review, we summarized the current understanding of EREG as an oncogene and discussed its oncogenic role in lung tumorigenesis and therapeutic resistance.

## 1. Introduction

Lung cancer has the highest mortality rate worldwide [1,2] and is histologically classified into small cell lung cancer (SCLC) and non-small cell lung cancer (NSCLC); the latter includes lung adenocarcinoma (LUAD), which accounts for 50–60% of all lung cancers [3,4]. Several druggable genetic alterations, including mutations in the kinase domain of the epidermal growth factor receptor (*EGFR*), *KRAS* G12C mutations, *BRAF* V600E mutations, *MET* exon14-skipping mutations, *HER2*/*ErbB2* exon 20 mutations, and fusions of *ALK*, *ROS1*, *RET*, and *NTRK*, have been identified in NSCLC (mainly in LUAD) [5,6]. Furthermore, immune checkpoint inhibitors, including antibodies to cytotoxic T lymphocyte-associated antigen 4 (CTLA-4), programed cell death-1 (PD-1), and programed cell death ligand 1 (PD-L1), have been reported to be clinically effective for patients with NSCLC [7,8]. Thus, the development of molecularly targeted therapies and immunotherapies has improved the survival of patients with NSCLC [9,10]. However, most patients with NSCLC receiving molecularly targeted therapies ultimately have progressive diseases because of “on-target” therapeutic resistance due to alterations in the targeted oncogenes or “off-target” therapeutic resistance due to the aberrant activation of the bypass and downstream signaling pathways [11]. For instance, HER2 overexpression is a mechanism of resistance to EGFR tyrosine kinase inhibitors (EGFR-TKIs) [12]. Another epidermal growth factor (EGF) family ligand, HER3, also plays a role in EGFR-TKI resistance through its own upregulation or interaction with *MET* amplification, thereby maintaining the antiapoptotic HER3/PI3K/AKT pathway to bypass EGFR downstream signaling [13]. Recent studies unraveled an essential role of AXL, a member of the Tyro3-Axl-Mer receptor tyrosine kinase family, in therapeutic resistance [14]. Treatment with EGFR-TKIs induces the activation of AXL, which in turn interacts with the EGFR and HER3 to maintain the survival of NSCLC cells [15]. AXL facilitates the upregulation of RAD18 and error-prone DNA polymerases to induce mutator phenotypes that confer adaptive resistance to EGFR-TKIs in NSCLC [16]. Moreover, the therapeutic efficacy of immune checkpoint inhibitors is not satisfactory given that the median overall survival for metastatic NSCLC patients treated with immune checkpoint inhibitors alone or combined with chemotherapy was only 14.4 months in a recent real-world study [17]. Diverse mechanisms underlie the resistance to immune checkpoint inhibitors, resulting in a lack of initial treatment response (primary resistance) and the development of resistance overtime (acquired resistance) [18]. Therefore, there is an urgent need to develop new treatment strategies to overcome drug resistance and to identify the predictive biomarkers of therapeutic responsiveness.

EGFR signal transduction regulates multiple cellular processes and plays a pivotal role in tumorigenesis [19]. The EGF/ErbB family of ligands includes EGF, transforming growth factor-α (TGF-α), amphiregulin (AREG), epigen (EPGN), heparin-binding EGF-like growth factor (HB-EGF), betacellulin (BTC), neuregulins 1–4, and epiregulin (EREG). They stimulate members of the EGFR family to activate downstream signaling pathways [20], and there is a cross-talk between these ligands; these EGF ligands can auto- and cross-induce one another [21,22]. EREG is encoded by the *EREG* gene on chromosome 4q13.3 and consists of 46 amino acids [23]. EREG was originally identified in the conditioned medium of mouse fibroblast-derived tumor cells [24]. EREG expression levels are extremely low in normal human tissues, including lung tissue [25]. EREG is proteolytically cleaved by members of the matrix metalloprotease (MMP) and a disintegrin and metalloprotease (ADAM) families, including ADAM17 [26,27,28]; the cleaved, mature EREG can activate EGFR signaling [29]. EREG can bind to the EGFR and ErbB4, activating the homodimers of the EGFR or ErbB4 and heterodimers of EGFR/ErbB2 and ErbB2/ErbB4 [20]. Interaction between EREG and these receptors activates multiple downstream pathways, including the MEK/ERK and PI3K/AKT pathways, thus regulating diverse cellular functions such as cell proliferation, differentiation, and migration [23,30,31]. EREG also plays a key role in angiogenesis, vascular remodeling, and wound repair during inflammation [23]. Several vasoactive G protein-coupled receptor agonists, such as angiotensin II, endothelin-1, and α-thrombin, activate ADAMs to release mature EREG, which in turn stimulates the proliferation of vascular smooth muscle cells through EGFR activation [32,33]. Additionally, the chemokine CX_3_CL1 induces EREG expression to promote proliferation of vascular smooth muscle cells through the activation of the MEK/ERK and PI3K/AKT pathways [34]. Notably, either EREG or epigen induces weaker and shorter lived EGFR dimers than EGF [35]. This weakened dimerization causes sustained EGFR signaling and promotes cell differentiation rather than cell proliferation, indicating the differential biological effects of the EGFR ligands on downstream signaling pathways. Thus, EREG has multiple roles in biological processes through regulating EGFR signaling.

Accumulating evidence has uncovered the overexpression of EREG in various human cancers, including NSCLC. EREG overexpression in tumors results in the oncogenic dysregulation of signaling pathways and induces malignant phenotypes such as enhanced cell proliferation, invasion, metastasis, angiogenesis, and resistance to regulated cell death [23]. Furthermore, increased EREG expression confers therapeutic resistance involving the epithelial–mesenchymal transition (EMT), cancer stemness, and tumor immune evasion. In this review, we summarized the current understanding of EREG as an oncogene and discussed its oncogenic role in lung tumorigenesis and therapeutic resistance.

## 2. Physiological Role of EREG in Human Airway Epithelial Cells

EREG regulates the proliferation and differentiation of human airway epithelial cells in an autocrine or paracrine manner. Various stimuli to the cells induce EREG expression through the ectodomain shedding of EREG by ADAM17. Human airway epithelial cells can be differentiated in a co-culture with pulmonary fibroblasts, where EREG is expressed during ErbB2 receptor phosphorylation [36]. Infection of human bronchial epithelial cells with rhinovirus or mycoplasma pneumoniae induces EREG expression, which in turn activates EGFR signaling [37,38]. EREG expression increases in response to compressive stress through EGFR signaling in human airway epithelial cells [39]. Compressive stress-induced EREG expression in murine tracheal epithelial cells (MTECs) is ablated by the ADAM17 inhibitor, TAPI-2, whereas the deletion of ADAM17 in MTECs decreases the EREG expression induced by compressive stress, indicating that ADAM17 is required for this process of EREG induction [40].

Previous studies have shown that exposure to various chemical carcinogens induces the EREG expression that is relevant to lung tumorigenesis. Cigarette smoking is a well-known risk factor for lung cancer [41], and carcinogens in cigarette smoke induce EREG upregulation in bronchial epithelial cells. The exposure of human bronchial epithelial cells to cigarette smoke extract upregulates EREG, which in turn promotes cell proliferation and protects against cigarette smoke-induced cytotoxicity [42,43]. The treatment of lung adenocarcinoma cells with benzo[a]pyrene, which is a potent carcinogen in cigarette smoke, increases EREG expression and promotes cell proliferation [44]. A previous study identified *EREG* as the most overexpressed gene in mouse lung tumors induced by vinyl carbamate, which is a carcinogen in cigarette smoke [45]. Some environmental carcinogens have also been shown to induce EREG expression in the lungs. EREG expression was increased in lung tumors induced by the environmental carcinogen of urethane in a murine experimental model [46]. Recently, Chen et al. [47] demonstrated that long-term exposure to 3-nitrobenzanthrone, which is a carcinogen in diesel exhaust, promotes malignant transformation in human bronchial epithelial cells through the EREG-mediated activation of the MEK/ERK and PI3K/AKT pathways. This is accompanied by the synergistic activation of IL-6 signaling, leading to lung tumorigenicity. These reports suggest that EREG upregulation following exposure to various chemical carcinogens contributes to lung tumor initiation.

## 3. Oncogenic Roles of EREG in Lung Cancer

Multiple lines of evidence have indicated the prognostic significance of EREG expression for many human cancers, including NSCLC (Table 1). We have previously reported that EREG was predominantly expressed in LUAD compared with lung squamous cell carcinoma and that elevated EREG expression was an independent prognostic marker in patients with LUAD [48,49]. Consistent with these findings, several studies have demonstrated that high *EREG* expression is associated with unfavorable overall survival for patients with LUAD [42,47,50]. Previous survival analyses using TCGA datasets have indicated that patients with LUAD with high EREG expression had significantly shorter survival rates than those with low EREG expression [42,50]. Another survival analysis using microarray datasets revealed an association between high EREG expression and unfavorable survival among patients with stage IA NSCLC [47]. Regarding the high EREG expression-related clinicopathological features, EREG is highly expressed in LUAD tumors with lymphatic permeation, vascular invasion, and pleural involvement [49]. In addition, EREG is predominantly expressed in LUAD tumors from patients who are older adults (≥70), males, and smokers [49]. Furthermore, elevated EREG expression correlates with lymph node metastasis in NSCLC [51] and the advanced stages of LUAD [42]. Collectively, these findings suggest that EREG overexpression contributes to an aggressive phenotype in NSCLC, especially in LUAD.

The oncogenic functions of EREG have been reported for various human cancers. Human recombinant EREG promotes cell proliferation in pancreatic [63] and bladder cancer [64] in a dose-dependent manner, whereas an *EREG* knockdown inhibits tumor growth in hepatocellular carcinoma [65], breast cancer [66], head and neck squamous cell carcinoma (HNSCC) [57], cervical cancer [54], glioma [67], and multiple myeloma [68]. Forced EREG expression enhances tumorigenicity in glioblastoma [58], salivary adenoid cystic carcinoma (SACC) [56], HNSCC [57], and esophageal cancer [59]. Moreover, EREG is upregulated in the metastatic lung tumors of breast cancer [69] and bladder cancer [70] in murine models of lung metastasis. EREG has also been identified as a liver metastasis-associated gene in colon cancer [71]. In *EGFR*-mutated NSCLC cells, EREG attenuation inhibits cell proliferation and invasion and induces apoptosis in *EGFR*-mutated NSCLC cells [51]. Similarly, we have previously demonstrated that a small interfering RNA-mediated EREG knockdown suppresses cell growth and induces apoptosis in NSCLC cells harboring *KRAS* mutations [48,49]. In a murine experimental model, tumors and inflammatory cells (macrophages and polymorphonuclear neutrophils) were reduced in mice lacking *Ereg*, indicating that EREG induces inflammation and lung tumor promotion [72]. These observations suggest that *EREG* functions as an oncogene in NSCLC cells. In contrast, EREG may be indispensable for SCLC tumorigenesis given that EREG expression is undetectable in most SCLC cell lines [48].

A recent study revealed that EREG is a core onco-immunological biomarker of cuproptosis [67], which is a molecular mechanism of regulated cell death [73]. Cuproptosis is induced by excessive copper, which binds to lipoylated dihydrolipoamide S-acetyltransferase (DLAT) [74]. The accumulated, insoluble DLAT causes cellular proteotoxic stress, thereby leading to cell death. Increasing evidence indicates the involvement of cuproptosis in the tumor progression and angiogenesis of lung cancer [75]. Recently, Zhou et al. [67] reported that EREG influences glioma cell growth by affecting the process of cuproptosis, which is accompanied by the EREG-mediated upregulation of FDX1, a core regulatory protein in cuproptosis. These findings suggest that EREG is involved in cuproptosis by regulating FDX1, although the role of EREG in cuproptosis remains to be elucidated for lung cancer.

## 4. EREG in Oncogene-Driven NSCLC

Several druggable molecular targets for NSCLC have been identified over the last two decades [5]. *KRAS* mutations are common driver mutations in NSCLC and are identified in 25–30% of patients with NSCLC [5]. Among the *KRAS* mutation subtypes, *KRAS* G12C mutations (approximately 40% of all *KRAS* mutations in NSCLC) are currently druggable drivers in NSCLC [76]. *EGFR* mutations in the tyrosine kinase domain are also common drivers in NSCLC, occurring in 10–40% of patients with NSCLC, with prevalence varying with ethnicity (highly observed for Asians) [6]. EGFR-TKIs are currently used as standard therapies for patients with NSCLC with *EGFR* mutations. Meanwhile, *BRAF* mutations are detected in 1.5–3.5% of NSCLC cases, and *BRAF* V600E mutations, which account for approximately half of all *BRAF* mutations in NSCLC, are druggable [77].

EGFR ligands appear to play roles in tumor promotion and drug resistance in oncogene-driven NSCLC. The elevated expression of TGF-α, which is a ligand binding to the EGFR, was associated with a poor prognosis for *EGFR*-mutated LUAD but not for wild-type LUAD, and TGF-α promoted the tumor growth of *EGFR*-mutated lung tumors in mice [78]. Moreover, TGF-α attenuated EGFR-TKI-mediated growth-inhibitory effects in NSCLC cells harboring *EGFR* mutations [79]. In *ALK* fusion-positive NSCLC cells, increased TGF-α expression confers resistance to an ALK tyrosine kinase inhibitor through the activation of EGFR bypass signaling [80]. HB-EGF, which is a ligand that binds to both EGFR and HER4, is abundantly expressed in *EGFR*-mutated NSCLC cells, whereas the HB-EGF inhibitor suppresses tumorigenesis [81]. Previous studies have suggested that EREG is upregulated by activating mutations in the *EGFR*, *KRAS*, and *BRAF*. Zhang et al. [51] reported that *EREG* mRNA expression was reduced through treatment with an EGFR-TKI or an MEK inhibitor in *EGFR*-mutant NSCLC cell in which EREG attenuation led to the inhibition of cell proliferation and invasion and induced apoptosis. In *EGFR*-mutant transgenic mice, lung tumors highly expressed EREG [82]. A previous study comparing gene expression profiles for colon cancer cells with or without oncogenic KRAS identified *EREG* to be significantly upregulated by the activated KRAS signaling pathway, and the forced expression of exogenous EREG enhanced in vivo tumorigenicity in a colon cancer xenograft model [83]. EREG is transcriptionally upregulated through MEK/ERK pathway activation in mutant KRAS-transformed prostate epithelial cells [84]. In addition, *EREG* mRNA expression is markedly increased in the neoplastic lesions of transgenic mice expressing mutant *KRAS* [85]. Consistent with these findings, we have previously reported that *EREG* is a transcriptional target of oncogenic KRAS signaling in both *KRAS*-mutant NSCLC cells and bronchial epithelial cells expressing ectopic mutant *KRAS* [86]. We also revealed that EREG is downregulated through the inhibition of the activities of oncogenic *EGFR* and *BRAF* in NSCLC cells [48,49]. In EREG-overexpressing NSCLC cells with *KRAS*, *EGFR,* or *BRAF* mutations, treatment with MEK and ERK inhibitors downregulates EREG expression, thereby suggesting that these oncogenic drivers induce EREG overexpression through MEK/ERK pathway activation [48,49]. Therefore, targeting EREG may be a therapeutic option for patients with NSCLC with oncogenic mutations in *KRAS*, the *EGFR*, or *BRAF*.

EREG is a substrate of ADAM17, which is a member of the ADAM family that is responsible for EREG release to the extracellular space in order to initiate the activation of EGFR signaling [28]. Previous studies have indicated the association between ADAM17 and oncogenic drivers for NSCLC. The oncogenic, KRAS-mediated activation of MEK/ERK signaling enhances ADAM17 activity in NSCLC cells [87,88]. In lung adenocarcinoma tissues, the number of phospho-ADAM17-positive cells was significantly increased in *KRAS*-mutated tumors than in the tumors with wild-type *KRAS* [89]. Additionally, ADAM17 is highly expressed in *EGFR*-mutated NSCLC cells, where an anti-ADAM17 antibody enhances EGFR-TKI-mediated growth inhibition, which is accompanied with reduced ERK phosphorylation [90]. Therefore, it is possible that such oncogenic drivers induce the EREG-mediated oncogenic activation of EGFR signaling through regulating ADAM17 activity in NSCLC.

## 5. EREG in the Tumor Microenvironment

Previous studies have elucidated the involvement of EREG in the pro-tumoral phenotype of the tumor microenvironment in which cancer-associated fibroblasts (CAFs) are the major source of EREG [23,91]. Neufert et al. [91] found that CAFs were the main producer of EREG in the tumor microenvironment of colitis-associated cancers and that EREG deficiency impaired colitis-associated tumor growth in mice, thereby indicating the tumor-promoting role of CAF-derived EREG. In a murine model of colonic adenoma, EREG was upregulated in colonic fibroblasts and recombinant EREG promoted the proliferation of colonic organoids [92]. Given that EREG is produced from fibroblasts [36] and macrophages in the lungs [93], EREG that is derived from stromal cells, such as fibroblasts and macrophages in the tumor microenvironment, may contribute to lung tumorigenesis. Xu et al. [42] demonstrated using a nitrosamine-induced mouse model that EREG expression was increased in the lung tumors of Mucin 1 (*MUC1*)-knockout mice, which was accompanied by the EGFR and AKT phosphorylation. They also showed that suppressing MUC1 enhanced EREG production in fibroblasts and lung cancer cells, whereas treatment with recombinant human EREG protected lung cancer cells from MUC1 inhibitor-induced cell death [42]. Their results suggest that MUC1 deficiency in fibroblasts and lung cancer cells increases the EREG production that activates EGFR signaling to compensate for the MUC1 loss, thereby promoting the activation of the EGFR and AKT pathways during lung carcinogenesis.

The EMT is a key cellular program that is implicated in tumor invasion and metastasis and confers stem cell properties and resistance to anticancer drugs [94,95]. The EMT also plays an essential role in tumor immunosuppression and immune evasion [96]. The TGF-β ligand is a key inducer of the EMT [97], and TGF-β signaling in the tumor microenvironment enhances cell proliferation and angiogenesis by activating the HB-EGF/IL-1β/EREG pathways [98]. YAP1, which is an upstream mediator of the EMT [99], upregulates EREG expression during intestinal stem cell proliferation and regeneration [100,101]. In a colon cancer xenograft model, EREG was expressed in both LGR5 (an intestinal stem cell marker)-positive and drug-resistant LGR5-negative cells [102], thereby indicating the involvement of EREG in cancer stemness. Furthermore, SACC cells overexpress EREG, which in turn promotes pulmonary metastasis by inducing the EMT and enhances angiogenesis by upregulating proangiogenic factors such as VEGFA, FGF-2, and IL-8 [56,103]. Notably, exosomes derived from SACC cells contain abundant EREG, and EREG-enriched exosomes promote the pulmonary metastasis of SACC in mice [56]. Cross-talk between the EREG and VEGFA ligands in the tumor microenvironment has also been found in a single cell transcriptome analysis uncovering that EREG mediates cuproptosis in glioma cells, where high cuproptosis activation scores were significantly enriched in VEGFA + malignant cells [67]. These results suggest that EREG affects tumor angiogenesis through regulating the VEGF signaling pathway.

## 6. EREG for Immune Evasion

Recent studies have indicated an immunosuppressive role of EREG. The upregulation of PD-L1 in cancer cells is a major mechanism of tumor immune evasion [104]. In glioma cells, an *EREG* knockdown reduced the expression of PD-L1 [67], whereas EREG was found to be functionally co-expressed with PD-L1 in LUAD [50]. Given that PD-L1 is upregulated by oncogenic *KRAS* mutations through the activation of the MEK/ERK pathway in NSCLC [105,106,107], there may be a link between EREG and PD-L1 in oncogenic KRAS-driven NSCLC. In fact, we observed a positive correlation between the mRNA expression of *EREG* and *CD274*, which encodes PD-L1, in LUADs (Figure 1A) and in *KRAS*-mutated LUADs (Figure 1B) by analyzing TCGA data.

The ecto-5′-nucleotidase CD73, which is encoded by the *NT5E* gene, is a cell-surface protein expressed in various types of cells, including immune cells, endothelial cells, CAFs, and tumor cells; it plays an essential role in tumor immune evasion and tumorigenesis [108,109,110]. CD73 functions to convert extracellular adenosine monophosphate (AMP) to adenosine and inorganic phosphate; extracellular adenosine then activates the adenosine receptor signaling that is responsible for immunosuppression [109]. CD73 is overexpressed in NSCLC, especially in NSCLC with *KRAS* mutations, *EGFR* mutations, and *ALK* fusions, through the activation of EGFR signaling [111,112,113]. In *EGFR*-mutated NSCLC cells, CD73 suppressed immune responses by inducing T regulatory cells (Tregs), and CD73 blockage inhibited tumor growth in an immune-competent murine model [111]. Notably, recent studies have suggested a relationship between CD73 and EREG in tumors through microRNA-330. In gastric tumors, CD73 expression is suppressed by miR-330-3p [114,115], whereas miR-330-3p directly targets and downregulates EREG expression to inhibit hepatocarcinoma progression [116]. Based on our previous microarray analysis identifying *NT5E* as the top-ranked upregulated gene by oncogenic *KRAS* in NSCLC cells [86], we herein investigated the expression levels of *EREG* and *NT5E* in LUADs and those with oncogenic *KRAS* mutations and identified a positive correlation between the expression of these genes (Figure 1C,D). Although further studies are needed to elucidate the roles of EREG in regulating PD-L1 and CD73, these findings suggest that EREG confers immune evasion in the lung tumor microenvironment.

**Figure 1 cancers-16-00710-f001:**
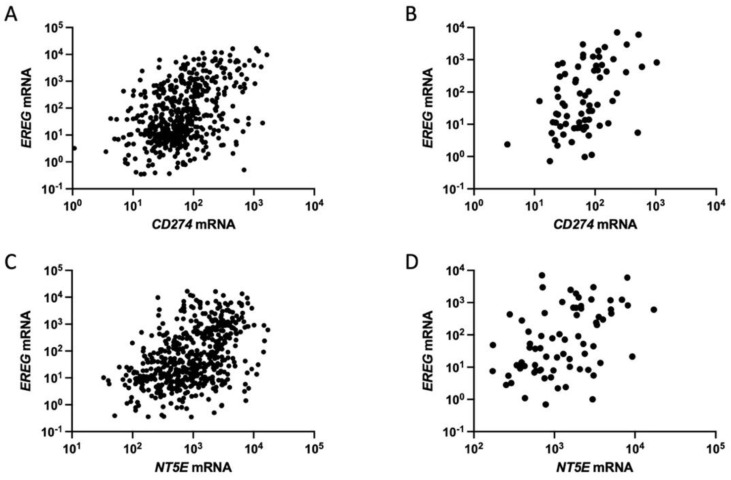
Positive correlation between *EREG* and *CD274* mRNA expression in (**A**) lung adenocarcinomas (N = 517, Spearman *r* = 0.41, *p* < 0.0001) and in (**B**) those with oncogenic *KRAS* mutations (N = 74, Spearman *r* = 0.46, *p* < 0.0001) and between *EREG* and *NT5E* mRNA expression in (**C**) lung adenocarcinomas (N = 517, Spearman *r* = 0.44, *p* < 0.0001) and in (**D**) those with oncogenic *KRAS* mutations (N = 74, Spearman *r* = 0.46, *p* < 0.0001) through TCGA database analysis (RNA Seq V2 RSEM dataset of TCGA, Firehose Legacy). *KRAS* mutations in codons 12, 13, and 61 were included in the analysis. The dataset was obtained from the website of cBioPortal for Cancer Genomics [117,118,119].

## 7. EREG and Resistance to Anticancer Drugs

While many anticancer drugs, such as molecularly targeted drugs, anti-angiogenic drugs, and immune checkpoint inhibitors, have been clinically used for the treatment of NSCLC, most patients with advanced diseases cannot be cured owing to the diverse mechanisms of resistance to anticancer therapies [4,11]. Aberrant EGFR signaling by upregulated EGFR ligands is one of the “off-target” resistance mechanisms causing the activation of the pathways downstream to or in parallel with the targeted molecules [120]. Considering the binding capacity of EREG to EGFR and ErbB4 to stimulate the homodimers of EGFR or ErbB4 and the heterodimers of EGFR/ErbB2 and ErbB2/ErbB4, it is plausible that the oncogenic upregulation of EREG results in the aberrant EGFR signaling that mediates the activation of the parallel bypass signaling pathways, thereby causing the “off-target” resistance to molecularly targeted drugs. In fact, *EGFR*-mutated NSCLC tumors highly express EREG [48,51,82], and treatment with EGFR-TKIs displayed limited antitumor activity against EREG-overexpressing lung tumors with *EGFR* mutations in a murine model [82]. In NSCLC patients who received the EGFR-TKI therapy, the disease control rate was lower in the EREG-high subgroup than in the EREG-low subgroup, and high EREG expression was associated with an unfavorable PFS [93].

The non-receptor protein tyrosine phosphatase SHP2 plays an essential role in the signal transduction that is mediated by receptor tyrosine kinases (RTKs) and T cell immune inhibitory signaling [121]. Recent studies have highlighted the significance of SHP2 in therapeutic resistance to molecularly targeted drugs and immune checkpoint inhibitors [122]. Drug sensitivity to the SHP inhibitor SHP099 for head and neck squamous cell carcinoma (HNSCC) is dependent on EREG [123]. HNSCC cell lines with low EREG expression are more sensitive to SHP099 than those with high EREG expression, and treatment with exogenous EREG sustains MEK/ERK and PI3K/AKT signaling and confers resistance to SHP099 in EREG-low HNSCC cells [123]. Thus, EREG overexpression appears to be responsible for the resistance to SHP2 inhibitors in cancer cells. This issue is particularly important given the ongoing clinical trials investigating the efficacy of several SHP2 inhibitors either as monotherapies or in combination with molecularly targeted drugs or immune checkpoint inhibitors for human cancers, including NSCLC [121].

Therapeutic resistance is associated with the EMT, cancer stemness, and the tumor microenvironment [94,124]. In patients with oral squamous cell carcinoma, EREG was highly expressed in CAFs compared with normal fibroblasts, and EREG-induced CAF activation promoted the EMT through IL-6 upregulation and JAK2-STAT3 pathway activation, thereby facilitating tumor growth and invasion [125]. EREG has been identified as a chemoresistance-associated gene based on RNA-seq data analysis of patients with lung cancer and NSCLC cell lines [52]. In this study, treatment with cisplatin or taxol increased EREG expression through the ERK1/2 phosphorylation in NSCLC cells, whereas EREG stimulation enhanced the viability of taxol-treated NSCLC cells with the elevated expression of stemness-associated genes, such as *Nanog* and *Sox2*. In addition, an EREG knockdown re-sensitized NSCLC cells to cisplatin and taxol, which was accompanied by the ERK1/2 dephosphorylation and the decreased expression of stemness-associated genes. Another study by Ma et al. [93] demonstrated that EREG induced resistance to EGFR-TKIs through the activation of EGFR/ErbB2 heterodimers in NSCLC cells. They reported that macrophages in the tumor microenvironment secrete EREG, which in turn leads to the formation of the EGFR/ErbB2 heterodimer and induces resistance to EGFR-TKIs through the activation of the PI3K/AKT pathway. Similarly, EREG production by stromal cells in the tumor microenvironment causes chemoresistance in prostate cancer [126]. Furthermore, chemotherapy-treated prostate cancer patients with EREG-high tumor stromata have worse disease-free survival than those with EREG-low stromata [126]. Therefore, it is likely that EREG derived from tumor cells and tumor stromal cells play a role in the resistance of NSCLC to anticancer therapies.

ADAM17 has been shown to drive therapeutic resistance in cancer [127], raising the possibility that ADAM17 may be complicit in EREG-mediated drug resistance. In EGFR-TKI-resistant NSCLC cells, an RNA interference-mediated ADAM17 knockdown recovers the sensitivity to the EGFR-TKI gefitinib through the dephosphorylation of EGFR [128]. Similarly, ADAM17 attenuation by an anti-ADAM17 antibody causes sensitization to EGFR-TKIs in NSCLC cells, which is accompanied by ERK inactivation [90]. Thus, it is possible that ADAM17 contributes to EREG-induced therapeutic resistance in lung cancer.

## 8. Targeting EREG/EGFR Pathways

The current understanding of the roles of EREG in lung tumorigenesis and therapeutic resistance accentuates the therapeutic potential of targeting the EREG/EGFR pathway in lung cancer. Anti-EGFR antibodies, such as cetuximab, with a higher affinity than ligands can competitively inhibit ligand binding to the EGFR extracellular domain, thus diminishing the EGFR ligand-mediated activation of EGFR signaling [129,130]. Treatment with cetuximab inhibited the EREG-mediated cell proliferation of glioma cells with abundant EREG expression [131]. In murine lung tumors bearing a mutated *EGFR*, add-on treatment with cetuximab increased the sensitivity to EGFR-TKIs [82]. In clinical application, high EREG expression was associated with a longer OS compared with low EREG expression in *RAS* wild-type colorectal cancer patients receiving anti-EGFR antibody therapies [132]. Therefore, anti-EGFR antibodies may be effective for the treatment of cancers in which EREG controls tumor growth and drug resistance in an autocrine or paracrine way.

Several anti-EREG antibodies have been developed and tested in preclinical and clinical studies. Treatment with an antihuman EREG antibody reduced the invasive abilities of *EREG*-overexpressing NSCLC cells harboring *EGFR* mutations [51]. In a metastatic murine model of colon cancer, treatment with an anti-EREG antibody induced antitumor activity against tumors derived from EREG-expressing LGR5-positive cells [102], thereby suggesting that targeting EREG is effective in defeating cancer stemness and chemoresistance. A humanized, anti-EREG monoclonal antibody with high affinity that targets cytotoxicity has been developed [133]. This anti-EREG antibody blocks the stimulation of EGFR signaling by EREG and not by EGF, and it inhibits the cell adhesion of EREG-expressing human colon cancer cells [134]. Recently, Wang et al. [126] reported that the viabilities of prostate cancer cells increased upon treatment with conditioned media from EREG-overexpressing prostate stromal cells and that treatment with an anti-EREG monoclonal antibody reduced this effect. This anti-EREG antibody also enhanced the tumor-suppressive effect of the anti-cancer agent mitoxantrone in a xenograft model composed of prostate cancer cells and EREG-overexpressing prostate stromal cells. When combined, these findings suggest that targeting the EREG/EGFR pathways could be a strategy for cancer therapy, especially when overcoming therapeutic resistance.

## 9. Conclusions and Future Directions

The oncogenic dysregulation of EGFR signal transduction is a common mechanism for driving lung tumors, and many anticancer drugs targeting the molecules in the EGFR pathway have been clinically used for the treatment of NSCLC. Considering that major driver mutations in NSCLC, such as *EGFR* and *KRAS* mutations, induce EREG overexpression and activate EGFR pathways through an autocrine loop mechanism, EREG could be an attractive therapeutic target for NSCLC. Recent studies have highlighted the role of EREG in the regulation of the EMT, angiogenesis, cancer stemness, and immune evasion in the tumor microenvironment. In addition, metalloproteinases, such as ADAM17, with the shedding activity of EREG could implicate EREG-induced lung tumorigenesis and therapeutic resistance. The schematic representation of the oncogenic role of EREG involving the EMT, angiogenesis, cancer stemness, immune evasion, and therapeutic resistance is shown in Figure 2. Thus, targeting EREG may enhance treatment efficacy and overcome resistance to anticancer therapies in NSCLC. Further investigations into the oncogenic roles of EREG will lead to the development of new treatment strategies for NSCLC.

## Figures and Tables

**Figure 2 cancers-16-00710-f002:**
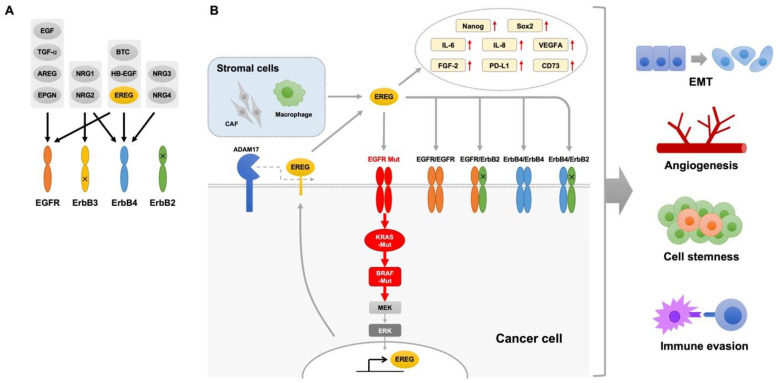
(**A**) Binding specificity of EGF family ligands: EGF, transforming growth factor-α (TGF-α), amphiregulin (AREG), epigen (EPGN), heparin-binding EGF-like growth factor (HB-EGF), betacellulin (BTC), neuregulins (NRGs), and epiregulin (EREG). EGF, TGF-α, AREG, and EPGN bind specifically to EGFR. BTC, HB-EGF, and EREG bind both to EGFR and ErbB4. NRG1 and NRG2 bind both to ErgB3 and ErbB4, whereas NRG3 and NRG4 bind only to ErbB4. (**B**) Schematic representation of the presumed role of epiregulin (EREG) involving the epithelial–mesenchymal transition (EMT), angiogenesis, cancer stemness, immune evasion, and therapeutic resistance. EREG can bind to EGFR and ErbB4, activates the homodimers of EGFR or ErbB4 and heterodimers of EGFR/ErbB2 and ErbB2/ErbB4, and activates multiple downstream pathways. EREG is proteolytically cleaved by a disintegrin and metalloprotease 17 (ADAM17) to release from the cells. Oncogenic mutations in *EGFR*, *KRAS*, and *BRAF* induce EREG overexpression in non-small cell lung cancer cells. Macrophages and cancer-associated fibroblasts (CAFs) are also sources of EREG in the tumor microenvironment. EREG upregulates various genes associated with angiogenesis, cancer stemness, and immune evasion.

**Table 1 cancers-16-00710-t001:** Previous studies showing the association between high EREG expression and unfavorable prognosis for human cancers.

	Source	No. of Subjects	Survival	Hazard Ratio (95% CI)	Reference
NSCLC (stage I–III)	Protein	356	OS	NA	[51]
NSCLC (stage IA)	mRNA	244	OS	1.62 (1.07–2.48)	[47]
NSCLC	mRNA	998	DFS & OS	NA	[52]
Lung adenocarcinoma	mRNA	119	DFS OS	2.33 (1.03–5.28) 8.71 (1.90–39.8)	[49]
	mRNA	462	OS	1.588 (NA)	[42]
	mRNA	520	OS	NA	[50]
Bladder cancer	mRNA	73	OS	NA	[53]
Cervical cancer	mRNA	304	OS	3.26 (2.03–5.21)	[54]
OSCC	mRNA	30	OS	NA	[55]
SACC	Protein	72	MFS & OS	NA	[56]
HNSCC	Protein	80	OS	NA	[57]
Glioblastoma	Protein	73	OS	NA	[58]
Esophageal cancer	Protein	120	OS	NA	[59]
Gastric cancer	mRNA	253	OS	NA	[60]
	Protein	550	OS	1.763 (1.235–2.480)	[61]
Pancreatic cancer	mRNA	NA	OS	NA	[62]

CI: confidence interval; DFS: disease-free survival; HNSCC: head and neck squamous cell carcinoma; MFS: metastasis-free survival; NA: not available; NSCLC: non-small cell lung cancer; OSCC: oral squamous cell carcinoma; OS: overall survival; SACC: salivary adenoid cystic carcinoma.

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
