# Peer review of "Role of Epiregulin in Lung Tumorigenesis and Therapeutic Resistance"

_cancers, 2024, doi:10.3390/cancers16040710_

Round 1
Reviewer 1 Report
Comments and Suggestions for Authors
In this review Aanuga et al revise the literature on the role of Epiregulin (EREG) in lung tumour progression ,
EREG is a member of the ErbB family of ligands that plays a role in cell proliferation, invasion, and angiogenesis and is involved in lung tumorigenesis and therapeutic resistance.
Although the topic is interesting and relevant I have several criticisms
The review is not completely focused and some topics are not well covered
The normal biological role of EREG and its signalling pathway need to be better described and paragraph 2 enlarged I accordingly
The effects of chemical carcinogens on EREG expression need to be enlarged and discussed in specific paragraph
The involvement of EREG in TME protumoral phenotype should be better discussed, i.e. with specific paragraphs focused on EREG and tumour progression , invasion and metastasis, stemness and drug resistance.
Finally, Authors should include a section discussing the potential therapeutic benefit of targeting EREG/EGFR pathways.
Author Response
We thank the reviewer for reviewing our manuscript and the thoughtful and constructive comments.
- The normal biological role of EREG and its signalling pathway need to be better described and paragraph 2 enlarged.
RE: We have revised second paragraph of the Introduction by adding the following sentences: "Interaction between EREG and these receptors activates multiple downstream pathways, including the MEK/ERK and PI3K/AKT pathways, thus regulating diverse cellular functions such as cell proliferation, differentiation, and migration [23,30,31]. EREG also plays a key role in angiogenesis, vascular remodeling, and wound repair during inflammation [23]. Several vasoactive G protein-coupled receptor agonists, such as angiotensin II, endothelin-1, and α-thrombin, activate ADAMs to release mature EREG, which in turn stimulates proliferation of vascular smooth muscle cells via EGFR activation [32,33]. Additionally, the chemokine CX3CL1 induces EREG expression to promote proliferation of vascular smooth muscle cells via activation of the MEK/ERK and PI3K/AKT pathways [34]." (page 2, lines 78-87) to describe normal biological roles of EREG and its signaling pathways.
- The effects of chemical carcinogens on EREG expression need to be enlarged and discussed in specific paragraph.
RE: We have revised the second paragraph of section 2 accordingly and added the following sentences: "A previous study identified EREGas the most overexpressed gene in mouse lung tumors induced by vinyl carbamate, a carcinogen in cigarette smoke [45]. Some environmental carcinogens have also been shown to induce EREG expression in the lungs. EREG expression was increased in lung tumors induced by the environmental carcinogen urethane in a murine experimental model [46]." (page 3, lines 122-126) to discuss the effects of several chemical carcinogens on EREG expression.
- The involvement of EREG in TME protumoral phenotype should be better discussed, i.e. with specific paragraphs focused on EREG and tumour progression, invasion and metastasis, stemness and drug resistance.
RE: We have enhanced the first paragraph of section 5 to account for the reviewer’s comment as follows: "Previous studies have elucidated the involvement of EREG in the pro-tumoral phenotype of the tumor microenvironment, in which cancer-associated fibroblasts (CAFs) are the major source of EREG [23,91]. Neufert, et al. [91] found that CAFs were the main producer of EREG in the tumor microenvironmentof colitis-associated cancers and that EREG deficiency impaired colitis-associated tumor growth in mice, indicating the tumor-promoting role of CAF-derived EREG. In a murine model of colonic adenoma, EREG was upregulated in colonic fibroblasts and recombinant EREG promoted proliferation of colonic organoids [92]." (page 6, lines 239-246) and " The TGF-β ligand is a key inducer of EMT [97], and TGF-β signaling in the tumor microenvironment enhances cell proliferation and angiogenesis by activating the HB-EGF/IL-1β/EREG pathways [98]." (page 6, lines 260-262). We here focused on EREG and tumor progression, invasion, and metastasis, while we believe that it is more reasonable to describe the involvement of EREG in stemness and drug resistance in the third paragraph of section 7 as follows: "In this study, treatment with cisplatin or taxol increased EREG expression via ERK1/2 phosphorylation in NSCLC cells, whereas EREG stimulation enhanced the viability of taxol-treated NSCLC cells with elevated expression of stemness-associated genes, such as Nanog and Sox2. In addition, EREG knockdown re-sensitized NSCLC cells to cisplatin and taxol, which was accompanied by ERK1/2 dephosphorylation and decreased expression of stemness-associated genes."(page 8, lines 350-355).
- Finally, Authors should include a section discussing the potential therapeutic benefit of targeting EREG/EGFR pathways.
RE: We have added section 8 entitled "Targeting EREG/EGFR Pathways" (page 8, line 373 to page 9, line 402) to discuss the potential therapeutic benefit of targeting EREG/EGFR pathways.

Reviewer 2 Report
Comments and Suggestions for Authors
This review article is considered informative and significant.
Authors explains that the median overall survival was reportedly only 8 months in a real-world study in section of introduction. However, the paper explained that overall survival was 8 months for first-line maintenance treatment. Therefore, the reference seems inadequate to present overall survival in patients with non-small cell lung cancer treated with immune checkpoint inhibitor therapy in clinical practice.
Author Response
We thank the reviewer for reviewing our manuscript and the helpful suggestion.
- Authors explains that the median overall survival was reportedly only 8 months in a real-world study in section of introduction. However, the paper explained that overall survival was 8 months for first-line maintenance treatment. Therefore, the reference seems inadequate to present overall survival in patients with non-small cell lung cancer treated with immune checkpoint inhibitor therapy in clinical practice.
RE: We have revised the sentence and added another reference by Hong L, et al. (Nat Commun, 2023) as follows: "Moreover, the therapeutic efficacy of immune checkpoint inhibitors is not satisfactory given that the median overall survival for metastatic NSCLC patients treated with immune checkpoint inhibitors alone or combined with chemotherapy was only 14.4 months in a recent real-world study [17]." (page 2, lines 57-60).

Reviewer 3 Report
Comments and Suggestions for Authors
Epiregulin (EREG), a member of the epidermal growth factor (EGF) family, plays a crucial role in tumorigenesis by binding to the epidermal growth factor receptor (EGFR) and ErbB4, thereby activating downstream pathways. Its aberrant expression, often triggered by activating mutations in EGFR, KRAS, and BRAF, contributes to aggressive phenotypes in non-small cell lung cancer (NSCLC). Recent studies emphasize EREG's impact on the tumor microenvironment, influencing processes such as epithelial-mesenchymal transition, angiogenesis, immune evasion, and resistance to therapy. This comprehensive review underscores EREG's role as an oncogene, providing valuable insights into its significance in lung tumorigenesis and therapeutic resistance, making it a crucial chapter in the field of lung cancer research. However, addressing the comments will be broaden the interest of the readers.
1. Authors have to include HER2, HER3, AXL (or TAM family members) and their role in acquired resistance to EGFR inhibitors in lung cancer. Also the regulation of how these receptors are regulating polymerases to mediate resistance. This can be in the introduction
2. EREG in Oncogene-Driven NSCLC : In this section authors have to briefly mention about other ligands that are EGFR binders and their role.
3. Authors have to mention about the cross talk of EREG and other ligands in the microenvironment
4. Authors have to introduce the advancement of therapy for EREG. Specifically using monoclonal antibodies and TRAPs for ligands
5. Authors have to discuss the role of EREG and immune evasion in a separate section.
6. It would be great if authors could include a scheme or a model.
Author Response
We thank the reviewer for reviewing our manuscript and the insightful comments that have helped us enhance the quality of the manuscript.
- Authors have to include HER2, HER3, AXL (or TAM family members) and their role in acquired resistance to EGFR inhibitors in lung cancer. Also, the regulation of how these receptors are regulating polymerases to mediate resistance. This can be in the introduction.
RE: We added sentences in Introduction as follows: "For instance, HER2 overexpression is a mechanism of resistance to EGFR-tyrosine kinase inhibitors (EGFR-TKIs) [12]. Another epidermal growth factor (EGF) family ligand, HER3, also plays a role in EGFR-TKI resistance through its own upregulation or interaction with MET amplification, thereby maintaining the antiapoptotic HER3/PI3K/AKT pathway to bypass EGFR downstream signaling [13]. Recent studies unraveled an essential role of AXL, a member of the Tyro3-Axl-Mer receptor tyrosine kinase family, in therapeutic resistance [14]. Treatment with EGFR-TKI induces activation of AXL, which in turn interacts with EGFR and HER3 to maintain survival of NSCLC cells [15]. AXL facilitates upregulation of RAD18 and error-prone DNA polymerases to induce mutator phenotypes that confer adaptive resistance to EGFR-TKI in NSCLC [16]." (page 2, lines 47-57) to describe the role of HER2, HER3, and AXL in the resistance to EGFR inhibitors and the mechanisms by which AXL regulates polymerases to mediate therapeutic resistance.
- EREG in Oncogene-Driven NSCLC: In this section, authors have to briefly mention about other ligands that are EGFR binders and their role.
RE: We added sentences in section 4 as follows: " EGFR ligands appear to play roles in tumor promotion and drug resistance in oncogene-driven NSCLC. Elevated expression of TGF-α, a ligand binding to EGFR, was associated with poor prognosis in EGFR-mutated LUAD, but not in wild-type LUAD, and TGF-α promoted tumor growth of EGFR-mutated lung tumors in mice [78]. Moreover, TGF-α attenuated EGFR-TKI-mediated growth-inhibitory effects in NSCLC cells harboring EGFR mutations [79]. In ALK fusion-positive NSCLC cells, increased TGF-α expression confers resistance to an ALK tyrosine kinase inhibitor through activation of EGFR bypass signaling [80]. HB-EGF, a ligand that binds to both EGFR and HER4, is abundantly expressed in EGFR-mutated NSCLC cells, whereas HB-EGF inhibitorsuppresses the tumorigenesis [81]." (page 5, lines 197-206) to describe the role of EGFR ligands other than EREG in oncogene-driven NSCLC.
- Authors have to mention about the cross talk of EREG and other ligands in the microenvironment.
RE: We have revised respective sentences in Introduction as follows " They stimulate members of the EGFR family to activate downstream signaling pathways [20], and there is a cross-talk among these ligands; these EGF ligands can auto- and cross-induce one another [21,22]." (page 2, lines 69-71) to describe the cross-talk among the EGF family of ligands. We have also added sentences in section 5 as follows: "The TGF-β ligand is a key inducer of EMT [97], and TGF-β signaling in the tumor microenvironment enhances cell proliferation and angiogenesis by activating the HB-EGF/IL-1β/EREG pathways [98]." (page 6, lines 260-262) to describe the cross-talk between EREG, HB-EGF, and TGF-β. We have further added sentences in section 5 as follows: " Cross-talk between EREG and VEGFA ligands in the tumor microenvironment has also been found in a single cell transcriptome analysis uncovering that EREG mediates cuproptosis in glioma cells, where high cuproptosis activation scores were significantly enriched in VEGFAmalignant cells [67]. These results suggest that EREG affects tumor angiogenesis via regulating the VEGF signaling pathway." (page 6, lines 271-275) to describe the cross-talk between EREG and VEGFA ligands.
- Authors have to introduce the advancement of therapy for EREG. Specifically using monoclonal antibodies and TRAPs for ligands.
RE: We have added section 8 entitled "Targeting EREG/EGFR Pathways" (page 8, lines 373 to page 9, line 402) to present advances in the therapy for EREG that include therapeutic strategies using anti-EREG monoclonal antibodies.
- Authors have to discuss the role of EREG and immune evasion in a separate section.
RE: We have discussed the role of EREG and immune evasion in a separate section by moving the related paragraphs from section 5 to section 6 entitled "EREG and Immune Evasion" (page 6, line 276 to page 7, line 305).
- It would be great if authors could include a scheme or a model.
RE: We have revised Figure 2 to describe the binding specificity of the EGF family of ligands (a) and to provide a scheme of the presumed role of EREG (b).

Round 2
Reviewer 1 Report
Comments and Suggestions for Authors
Authors have improved the manuscript acconrding with the suggestions. I reccomend publication of the manuscript
Reviewer 3 Report
Comments and Suggestions for Authors
The authors have addressed all the comments.